# Simple and Effective Input Reformulations for Translation

**Brian Yu, Hansen Lillemark, Kurt Keutzer**
University of California, Berkeley
Berkeley Artificial Intelligence Research (BAIR)
`{bri25yu,hlillemark,keutzer}@berkeley.edu`

## Abstract

Foundation language models learn from their finetuning input context in different ways. In this paper, we reformulate inputs during finetuning for challenging translation tasks, leveraging model strengths from pretraining in novel ways to improve downstream performance. These reformulations are simple data level modifications, require no additional collection of training data or modification of data at inference time. They can be applied either on single language pair translation tasks or massively multilingual translation tasks. Experiments with these techniques demonstrate significant performance improvements up to **3.5 chrF++ on the Flores200 translation benchmark**. We hope our research accessibly improves finetuning data efficiency, enabling more effective training to scalably improve state-of-the-art performance. Our code is released here.

## 1  Introduction

Foundation language models (FLMs) are powerful and task-agnostic models. They are pretrained on language understanding objectives, enabling strong performance on downstream language tasks (Brown et al., 2020; Shoeybi et al., 2020; Xue et al., 2021; Hoffmann et al., 2022; Chowdhery et al., 2022; Zhang et al., 2022a; Chung et al., 2022; Workshop, 2023; Touvron et al., 2023). FLMs are then either prompted or finetuned for downstream use.

In this paper, we present three different data efficient techniques for improving translation performance, applied to the multilingual FLM mT5 during finetuning (Xue et al., 2021). In our first approach, we train mT5 on a Classical Tibetan to English (tib2eng) translation task. mT5 struggles heavily in the initial training steps. Thus, for the first 20% of finetuning, we apply the "**P**artial **O**utput **E**nglish **S**caffold" or POSE reformulation, shown in Figure 1. Tib2eng translation examples consist of a Classical Tibetan source and English

target translation pair. POSE simply appends a prefix of the target English output to the Classical Tibetan input. We see qualitative improvements in the variance of the training curves. When evaluated on the same test set with no reformulations, POSE significantly increases overall translation performance compared to the direct finetuning baseline, up to **10.3% / 2.8 BLEU**.

The POSE setup had many adjustable hyperperameters relating to task difficulty, task curriculum, and substring selection for scaffolding. We find that input reformulation setups should consist of 20% less informative examples, and 80% harder and more informative examples. More ablation details can be found below.

Second, we approach the massively multilingual Flores200 translation benchmark (NLLB-Team et al., 2022). mT5 does not struggle in the initial steps of finetuning on Flores200 in the same way it did on tib2eng. Even so, we begin by replicating the tib2eng POSE setup on Flores200 by appending a partial output of the target translation to the input translation. As expected, this setup matched but did not improve upon the baseline performance.

The Flores200 benchmark consists of parallel examples of the same sentence in different languages. In our second approach, we extend the tib2eng POSE reformulation to create the "**Par**allel **S**caffold in **E**nglish" or ParSE reformulation, shown in Figure 1. ParSE appends the corresponding full parallel English translation (provided by Flores200) to the input. Following the tib2eng setup, we use a data mix of 20% baseline (less informative) and 80% ParSE (more informative) examples. ParSE significantly improves translation performance, up to **17.2% / 3.5 chrF++**.

We postulate that POSE and ParSE improve translation performance in part because they enable mT5 to attend to an in-distribution pretrain language with strong monolingual performance. In our third approach, we explore the efficacy of par-

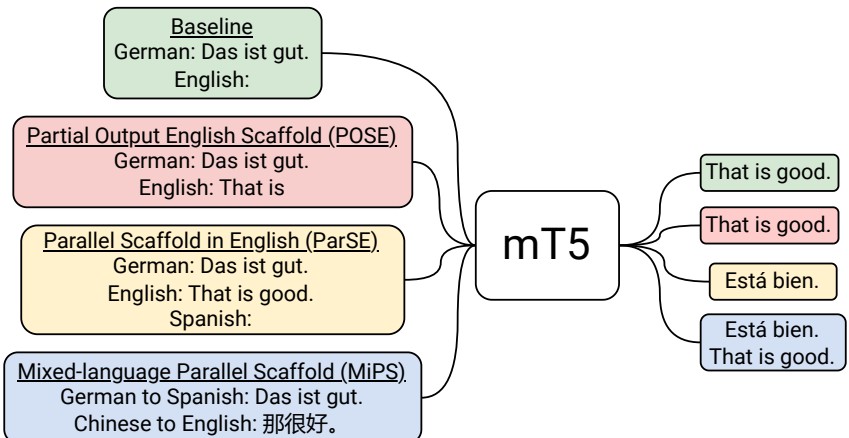

Figure 1: Task reformulations. **Baseline**: a direct translation pair. **POSE**: append a prefix of the target translation to the input translation. **ParSE**: append a parallel English translation to the input translation. **MiPS**: append a different parallel translation to both the input and output.

allel scaffolding that does not require strong monolingual performance using the "**Mi**xed-language **P**arallel **S**caffold" or MiPS reformulation, shown in Figure 1. MiPS appends a different parallel translation to both the input and output for a total of 4 distinct languages per input. Again, we use a data mix of 20% baseline and 80% MiPS examples. MiPS also improves translation performance, up to **9.1% / 1.6 chrF++**. Scaffolding with the strongest performing pretraining language (ParSE) outperforms scaffolding with a mix of other languages (MiPS).

Finally, we perform analysis on the languages in the translation set. Using a balanced dataset like Flores200 allows mT5 to partially overcome pretraining dataset size biases. Naturally, translating *into lower resource* languages is more difficult than translating *into higher resource* languages, but we find that the ParSE and MiPS reformulations improve translation into all languages across the board, rather than disproportionately improving performance on high resource languages.

In summary, we propose input reformulations on translation tasks. These reformulations require no additional data, have few hyperparameters, and are simple to implement. When finetuning on a single language pair translation task, if the target output language is in the model's pretraining dataset distribution, the POSE reformulation can be applied. When translating between multiple language pairs, the ParSE reformulation can be applied to the strongest performing pretraining language.

## 2    Related work

Our work can be viewed as a data efficiency technique for translation. Past works in translation have explored data augmentation (Sennrich et al., 2016; Fadaee et al., 2017), sample re-weighting (Shu et al., 2019; Ren et al., 2019; Gu et al., 2018), and curriculum learning (Kocmi and Bojar, 2017; Zhang et al., 2018; Platanios et al., 2019; Zhang et al., 2019; NLLB-Team et al., 2022). These approaches vary in effectiveness, are not generalizable, and introduce complexity into the training process. Curriculum learning approaches in particular are typically complicated and unsuccessful, because they are designed using intuition on how *humans* treat inputs, which may differ from how *models* treat inputs. In contrast, our input reformulations are simple and can be directly applied to any sequence-to-sequence task.

Previous work has explored prompting a frozen language model using manually curated prompts (Brown et al., 2020; Touvron et al., 2023; Petroni et al., 2019). Results are typically sensitive to the exact prompt used. This technique cannot be applied to larger corpora because it is limited by the number of examples that can feasibly fit into a single input context. Other works have explored finetuning with a fixed prompt without leveraging the target output as a part of the input (Radford et al., 2018, 2019; Dong et al., 2019; Devlin et al., 2019; Lewis et al., 2019; Sun et al., 2019; Liu et al., 2019; Clark et al., 2020; Yang et al., 2020; Raffel et al., 2020; Gao et al., 2021; Schick and Schütze, 2021; au2 et al., 2021; Xue et al., 2021; He et al., 2021; Taori et al., 2023).

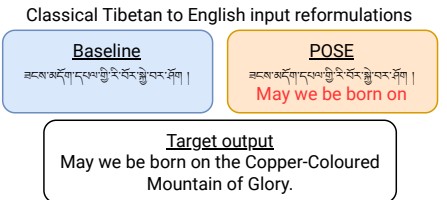

Figure 2: POSE reformulation applied to the tib2eng translation task. Changes are highlighted in **red**.

Following the success of fixed prompt techniques, other works proposed prompt tuning setups (Shin et al., 2020; Schick et al., 2020; Li and Liang, 2021; Hambardzumyan et al., 2021; Lester et al., 2021; Zhong et al., 2021b; Wallace et al., 2021; Haviv et al., 2021; Jiang et al., 2020; Chen et al., 2022; Qin and Eisner, 2021; Liu et al., 2021; Han et al., 2021; Zhong et al., 2021a; Lu et al., 2022; Ben-David et al., 2022; Wang et al., 2022a; Zhou et al., 2023b). These prompt tuning setups were typically used in the context of compute efficiency: training a smaller number of prompt-related parameters to input into a larger frozen language model. These setups are an orthogonal improvement to our proposed input reformulations.

Previous approaches also investigated dataset improvements for better downstream task performance. These approaches gathered additional data for model training to augment the model's input context (Chung et al., 2022; Wei et al., 2023; Wang et al., 2023a; Iyer et al., 2023; Min et al., 2022; Wei et al., 2022; Wang et al., 2022b; Gu et al., 2023; Wang et al., 2023b; Zhang et al., 2022b; Press et al., 2023; Zhou et al., 2023a). They require large, specific, and high quality datasets to be collected. On the other hand, our input reformulations require no additional data.

Overall, our approach differs from previously explored approaches by avoiding prompts and leveraging the target output as a part of the input reformulation. Our input reformulations are a data-level change that can be easily applied to any training setup.

## 3 Experiments on a difficult single language pair translation task

### 3.1 Setup

We perform experiments on a Classical Tibetan to English (tib2eng) dataset. Critically, Classical Tibetan is not found in mT5's pretraining dataset, while English is. As a result, the tib2eng dataset

is challenging for mT5. Additionally, mT5's tokenizer was not trained on Tibetan. We use mT5's current tokenizer and use the byte-level fallback capabilities of the underlying SentencePiece tokenizer to encode unknown tokens (Xue et al., 2021). We use the BLEU metric (Papineni et al., 2002) for evaluation.

The dataset consists of 450k train, 5k validation, and 5k test translation pairs. The tokenized Tibetan inputs are mean 72 and median 51 tokens long; we use a maximum sequence length of 256. We train for 10k steps and a batch size of 512 translation pairs (about 35k tokens per batch, about 350M tokens total), equivalent to 11 epochs. We use the AdamW (Loshchilov and Hutter, 2019) optimizer with parameters $\beta_1 = 0.9$, $\beta_2 = 0.999$, and weight decay 0. We use a constant learning rate schedule with no warmup. The models converge successfully under this data compute budget. We ablate over learning rates in {1e-3, 2e-3, 3e-3} for 600M and 1B parameter models (the default finetuning learning rate for mT5 is 1e-3 (Xue et al., 2021)) and {3e-4, 5e-4, 1e-3} for 3B parameter models, where we found lower learning rates to be empirically better.

We perform evaluation on the models and save checkpoints every 200 steps, for a total of 50 evaluations, and we use the highest scoring checkpoint for all results. Models were trained on GPU nodes of either 8 NVIDIA A5000 24GB GPUs or 8 NVIDIA A6000 48GB GPUs. The typical train time varied from 8 hours for the smallest models to 80 hours for the largest. We leverage the Deepspeed library https://www.deepspeed.ai/ for training in the half precision bf16, as well as for effective multi-GPU training.

In all the following results tables, we report the highest test set BLEU scores and standard deviation (std) values over learning rates.

### 3.2 Motivation

We begin by training baseline mT5 models on the tib2eng dataset. The resulting training curves are shown in Figure 3 with the blue colored curves. Clearly, mT5 struggles in the first 2000 steps or 20% of the training steps. With the intuition of reducing task difficulty, we design an easier task reformulation to apply only in the first 20% of training. First, we select a prefix from the target English translation. The length of this prefix is uniformly randomly chosen over the full length of the English

Table 1: Task difficulty experiment results on mT5 600M.

| Difficulty ↓ | % reform | BLEU | Std |
|---|---|---|---|
| Least difficult | 100% | 21.1 | 0.29 |
| | 50% | 23.9 | 0.05 |
| | **20%** | **24.6** | **0.26** |
| Most difficult | 0% | 23.5 | 1.64 |

Table 2: Curriculum experiment results on mT5 600M.

| Setup | BLEU | Std |
|---|---|---|
| Baseline | 23.5 | 1.64 |
| POSE | **24.6** | **0.26** |
| (Curriculum 1) | 17.4 | 0.85 |
| (Curriculum 2) | 24.9 | 0.74 |
| (Curriculum 3) | 24.7 | 2.50 |

Table 3: Prefix+suffix experiment results on mT5 600M.

| Substring | % reform | BLEU | Std |
|---|---|---|---|
| Baseline | 0% | 23.5 | 1.64 |
| Prefix | **20%** | **24.6** | **0.26** |
| Prefix+suffix | 12% | 24.8 | 0.55 |
| | 20% | 24.5 | 0.90 |
| | 40% | 24.0 | 0.12 |

translation. Then, we append this English prefix to the Classical Tibetan translation input. Intuitively, we "scaffold" the Classical Tibetan input with a partial English translation. We use a partial prefix of the English translation so the model doesn't degenerate into simply outputting all the English in the input. We name this reformulation "**P**artial **O**utput **S**caffold **E**nglish" or POSE. An example of POSE is found in Figure 2. The next 4 subsections cover ablations over the finetuning reformulation setup. For direct results on the POSE task, which ended up being the most successful, see section 3.7.

### 3.3 Modulating task difficulty

The POSE reformulation is easier than the baseline task. In order to modulate task difficulty, we ablate over different amounts of training examples that use this reformulation: 0% (baseline), 20%, 50%, and 100% (all reformulated).

Results are found in Table 1. The best condition involves reformulating the first 20% of training examples, achieving 24.6 BLEU, 1.3 BLEU higher than the baseline. We hypothesize that making the task too easy e.g. 50% or 100% reformulated makes the task less informative, which hurts downstream performance. All of the reformulated runs have low variance across the learning rates, suggesting that models are better conditioned while training on easier tasks.

### 3.4 Optimizing the curriculum

We attempt to optimize the curriculum using human intuition in 3 setups. (**Curriculum 1**): Instead

of reformulating only the first 20% of training examples (i.e. all examples in the first 2000 steps), we rigidly add 100% of the output to the input at the beginning of training, and linearly scale down to 0% added at the end of training. (**Curriculum 2**): Instead of reformulating 100% of training examples in the first 2000 steps, we reformulate 80% of the inputs for the first 2000 steps, linearly scale down from 80% reformulated to 40% reformulated for the next 4000 steps, and reformulate no examples for the last 4000 steps. (**Curriculum 3**): Instead of using uniformly random length prefixes for the first 20% of training examples, we rigidly add 100% of the output to the input and linearly scale down to 0% at the end of 2000 steps.

Results are found in Table 2. Even though these setups have merit using human intuition, mT5 performs markedly worse on all of them in either performance, stability, or both. The best performing runs perform better than POSE, but at the cost of stability.

### 3.5 Modulating scaffold substring

Rather than using just a prefix of the target English output, we experiment with setups that append both a portion of the target English prefix and a portion of the target English suffix ("prefix+suffix" reformulation). The total selected length remains the same for the prefix+suffix experiments. The prefix+suffix input reformulation is still in natural language, but using different pieces of the target output. Additionally, we perform a more fine-grained sweep over how many initial training examples are reformulated.

Results are found in Table 3. The prefix+suffix reformulation performs better and is less varied than the baseline, but performs worse than the prefix-only reformulation. We hypothesize that the prefix-only reformulation performs the best because it is the simplest. Over different amounts of initial training examples reformulated, 12% reformulated had the best raw performance, closely

Table 4: Matching pretraining experiment results on mT5 600M with masking.

| Setup | BLEU | Std |
|---|---|---|
| Baseline | 23.5 | 1.64 |
| POSE | **24.6** | **0.26** |
| (Mask 1) | 24.9 | 0.35 |
| (Mask 2) | 23.6 | 0.20 |
| (Mask 3) | 23.0 | 0.15 |
| (Mask 4) | 23.4 | 0.04 |

followed by 20%. We chose to stick with the 20% experiment due to the lower variance.

### 3.6 Matching the pretraining task

We hypothesize that matching the pretraining task smooths performance similar to the POSE reformulation. We experiment on 4 masking setups: **(Mask 1)** mask in the first 20% of finetuning steps with p=0.1; **(Mask 2)** mask in the last 20% of finetuning steps with p=0.1; **(Mask 3)** mask in the last 50% of finetuning steps with p=0.25; and **(Mask 4)** span-mask in the last 50% of finetuning steps with p=0.25. Results are found in Table 4. Masking setups have less variance compared to the baseline or previous best setup, most likely because they are closer to the pretraining task distribution. Setup (Mask 1) performs better than the POSE reformulation with slightly higher variance. However, we retain the POSE reformulation as the best because it is simpler than setup (Mask 1). The other masking setups (Mask 2), (Mask 3), and (Mask 4) result in lower performance, most likely because the task is less informative to the actual downstream translation task.

### 3.7 Final results and comparison to state-of-the-art

We select the best setup based on stability, simplicity, and performance. The best reformulation was still the original POSE reformulation. We compare performance of the baseline and POSE mT5 conditions with the state-of-the-art translation model NLLB (NLLB-Team et al., 2022). Because NLLB is a translation-only model, our input reformulations cannot be applied to it. NLLB's encoded input lengths are mean 26 / median 19 tokens. For NLLB, We ablate over learning rates in {3e-4, 5e-4, 1e-3}. For the NLLB tib2eng baseline, we use a linear warmup of 1000 steps, 10% of the total number of updates, with constant learning rate afterwards. The final results comparing the finetuning of mT5

Table 5: **Main results** on the tib2eng translation task for mT5. Values shown are test set BLEU scores. The difference shown is the improvement gained by using the input finetuning reformulations. The NLLB column is the test set BLEU score for the corresponding sized NLLB model.

| Params | NLLB | Baseline | POSE | Diff |
|---|---|---|---|---|
| 600M | *29.3* | 23.5 | 24.6 | **+1.1** |
| 1B | *32.3* | 27.2 | 28.3 | **+1.1** |
| 3B | *34.4* | 27.3 | 30.1 | **+2.8** |

baseline, mT5 POSE, and NLLB on the tib2eng task are shown in Table 5 and Figure 3.

The POSE reformulation stabilizes training and improves performance, with the largest mT5 3B model exceeding the performance of NLLB 600M. Additionally, while the baseline runs have converged, the mT5 POSE and NLLB models could be trained further for higher performance. NLLB has strong performance on this finetuning task despite not being trained on Classical Tibetan. This is because NLLB was trained on modern Tibetan, similar to classical Tibetan, and because NLLB is a translation-only model with a strong translation inductive prior. Our finetuning paradigm begins to bridge the gap between FLMs such as mT5, and task-specific translation-only models such as NLLB.

## 4 Experiments on a massively multilingual translation task

### 4.1 Setup

The Flores200 dataset consists of around 3,000 parallel sentences in 204 different languages, meaning each sentence is translated into all 204 languages with high fidelity (NLLB-Team et al., 2022; Goyal et al., 2021; Guzmán et al., 2019). This dataset is challenging for mT5 not only because of the sheer number of languages, but also because mT5 was not pretrained on over half of the languages present in the dataset. The Flores200 dataset is purported for evaluation with a separate, partially parallel train set, but the fully parallel nature of the Flores200 dataset enables interesting reformulations for finetuning. We take translation pairs from the Flores200 dev set as our training set, and translation pairs from the devtest set as our validation and test sets.

Our reformulated Flores200 dataset for training consists of 20M train, 5k validation, and 10k test

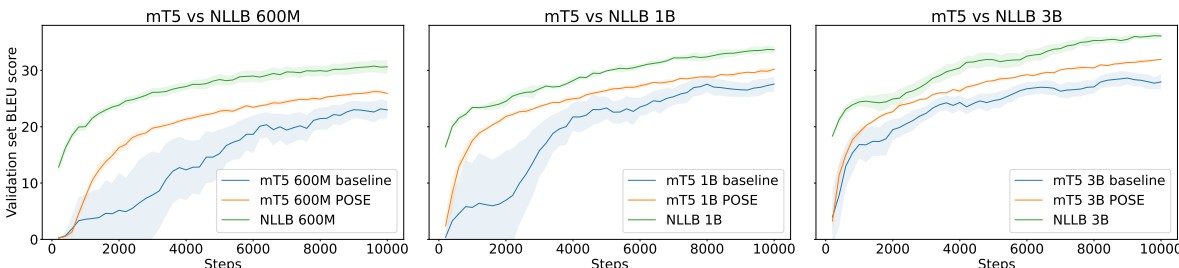

Figure 3: Tib2eng translation task reformulation experiment results. These results compare the **mT5 baseline (blue)**, **mT5 POSE (orange)**, and the **NLLB (green)** experimental configurations. The solid lines and shaded areas are the mean and variance over learning rates, respectively. Left: 600M. Center: 1B. Right: 3B.

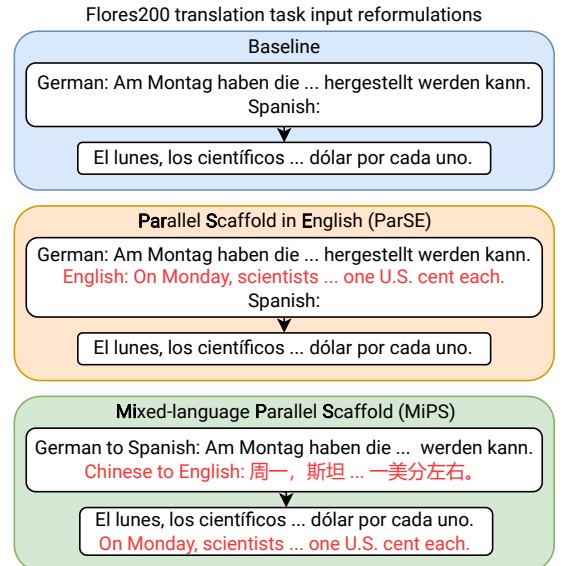

Figure 4: Examples of the ParSE and MiPS input reformulations applied to the Flores200 translation task. The changes to the original input are highlighted in **red**.

translation pairs. Following the tokenization setup for the tib2eng task, mT5's tokenizer yields inputs of mean 52 / median 46 tokens and we use a max sequence length of 256. We follow the NLLB team and perform evaluation on the Flores200 task using the chrF++ metric (Popović, 2015) with the xx-yy condition to present the final average score across languages (NLLB-Team et al., 2022). We ablate over the learning rates {1e-4, 2e-4, 3e-4}, where we found lower learning rates to be empirically better. We train for 10k steps with a batch size of 2048 examples (approximately 105,000 tokens).

## 4.2 Designing task reformulations

For the tib2eng task, we designed POSE to mitigate mT5's struggles early in finetuning. mT5 does not struggle in the same manner on Flores200. Even so, we begin by replicating the tib2eng POSE setup on Flores200 by appending a partial output of the target translation to the input translation. We experiment on mT5 300M. The baseline model achieves 16.8 validation set chrF++ and the reformulated model achieves 16.7 validation set chrF++. As expected, this setup matched but did not improve upon the baseline performance.

mT5 has strong English performance because it was pretrained on orders of magnitude more English data than other languages. So, we look to leverage this strong capability in an input reformulation. The Flores200 benchmark consists of parallel examples of the same sentence in different languages. We extend the tib2eng POSE reformulation to the "**Par**allel **S**caffold in **E**nglish" or ParSE reformulation. ParSE appends a full parallel English translation to the input translation. For the ParSE setup, we provide the intuition that English is used as a pivot language between the two other languages.

We explore the efficacy of parallel scaffolding without using English using the "**Mi**xed-language **P**arallel **S**caffold" or MiPS reformulation. MiPS appends a different parallel translation to both the input and output for a total of 4 distinct language translations per input. For simplicity, we use any combination of languages in Flores200, regardless if they're in or out of mT5's pretraining distribution. Examples of the ParSE and MiPS reformulations are shown in Figures 1 and 4.

For both the ParSE and MiPS reformulations, we follow the tib2eng setup and a data mix of 20% baseline (less informative) and 80% reformulated (more informative) examples. We use a data mix rather than reformulating the last 80% of training examples to further simplify setup and expose the

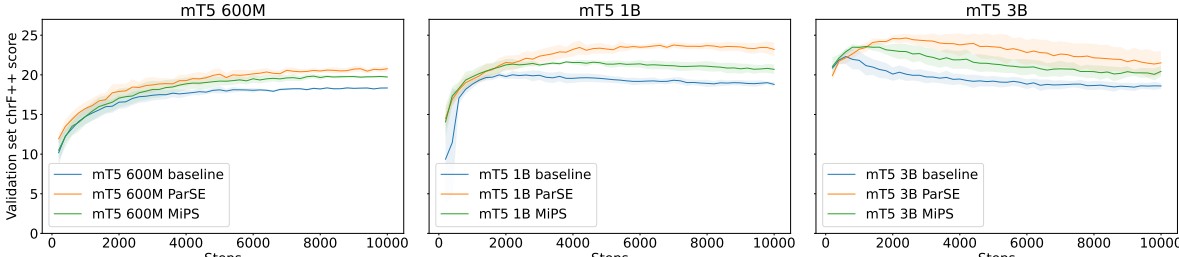

Figure 5: Flores200 translation task reformulation experiment results. These results compare the **mT5 baseline (blue)**, **mT5 ParSE (orange)**, and **mT5 MiPS (green)** experimental configurations. The solid lines and shaded areas are the mean and variance over learning rates, respectively. Left: 600M. Center: 1B. Right: 3B.

Table 6: Results on the Flores200 translation task for mT5. Values shown are test set chrF++ scores. The NLLB column is the task performance of a corresponding size NLLB model. For the NLLB score, we use the 200 xx-yy chrF++ scores listed here.

| Params | NLLB | Baseline | ParSE | MiPS |
|--------|------|----------|-------|------|
| 600M | *39.5* | 17.6 | **20.7** | 19.2 |
| 1B | *41.5* | 20.3 | **23.8** | 21.6 |
| 3B | *41.8* | 23.2 | **25.1** | 23.6 |

model to the input reformulations early in training. The input reformulations use up to twice the number of examples per input so we reduce the per-step batch size by a factor of two from 2048 to 1024 in order to hold the data and compute budgets constant across experiments.

### 4.3 Results

Our results are presented in Figure 5 and Table 6. We observe positive effects on performance similar to the tib2eng results. For the ParSE reformulation, the model learns slightly slower initially, but learns much more over the course of training. For the MiPS reformulation, the model learns faster and better than the baseline. Clearly, our input reformulation scheme improves performance, beyond just relying on strong English performance. We hypothesize that both tasks successfully improve performance, in part because they allow for direct attention between the input context in different languages, aligning representations across languages.

Interestingly, the ParSE reformulation performs the best, but also has the highest variance over the learning rates. The need for lower learning rates typically indicates poor conditioning, so the input task is likely more ill-conditioned than the baseline. One possible explanation is that mT5 is learning

the languages in Flores200 that were not present in its training set.

### 4.4 Analysis on mT5's pretraining dataset and Flores200

Flores200 contains 204 languages, while mT5 was only pretrained on 95 of them. We perform additional analysis on how being pretrained on a language affects the post-finetuning performance on Flores200, as well as how the pretraining data size for a specific language affects performance, shown in Figure 6. Translating from a language in the pretraining set into other languages is more difficult than translating from other languages into a language in the pretraining set. This is most likely because decoding into lower-resource languages is more difficult than encoding them.

When translating from a language in the pretraining set into other languages, pretraining data size is slightly correlated with better performance. However, this correlation is small considering the large range of dataset sizes. The ParSE and MiPS reformulations improve performance across the board, not depending on pretraining data size. Using a balanced finetuning dataset like Flores200 helps mitigate some of the language frequency related pretraining biases of mT5.

The performance improvement using ParSE when translating from English into other languages is much more pronounced. This can be seen visually in Figure 6 for the rightmost datapoint in each plot in the top row. The corresponding numbers in Table 7 for 3B models shows the increase for from-English is 6.3 chrF++. This makes intuitive sense since the model has seen significantly more English in the input during finetuning.

We break down the performance of different model sizes and reformulation setups in Table 7.

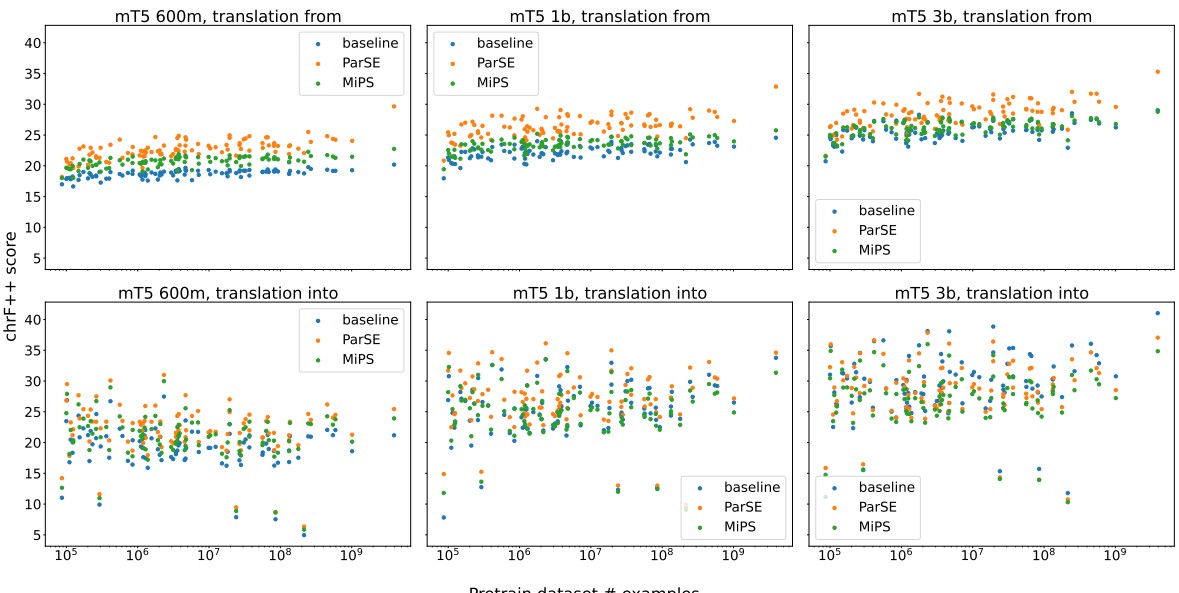

Figure 6: Pretraining dataset sizes and Flores200 finetuning performance. The first row represents translation from a language in the pretraining set into other languages, including those not in the pretraining set. The second row represents translation from other languages into a language present in the pretraining set. Each dot represents one language and the value in the graph represents the corresponding chrF++ test set score for that language and model. Points shown only cover languages present in the mT5 pretraining set. The point corresponding to English is the rightmost point on all the graphs. Dataset sizes are calculated using the number of examples of each language present in the mC4 dataset. Dataset sizes range from 100k to 1B examples.

Interestingly, the ParSE and MiPS reformulations improve performance involving lower-resource languages, sometimes at a slight cost to performance on higher resource languages. For example, the 3B baseline and ParSE conditions perform about the same when translating from languages in the pretrain dataset to other languages in the pretrain dataset. The ParSE condition performs 1.3 chrF++ worse than the baseline when translating from out-pretrain to in-pretrain languages. However, the ParSE condition performs significantly better than the baseline condition on the in-out and out-out language pairs, with chrF++ improvements of 5.3 and 3.6 respectively. Explanations for this requires further targeted experimental investigations.

## 5  Conclusion

We have explored how FLMs learn from their input contexts. We provide two separate techniques that can be applied to any translation use case. For the case of a single language pair translation task, we recommend POSE. For the case of a multi-language pair translation task, we recommend ParSE and MiPS. For challenging translation tasks, our scaffolding reformulations produce bet-

ter conditioned training curves and significantly better performance. These input reformulations are simple to understand and implement, robust over hyperparameters, general to translation tasks, and effective. We hope our technique is used to accessibly improve data efficiency on translation tasks.

## Limitations

Our proposed technique has only been applied to two challenging translation tasks, where the input and output are both information rich and sequential in nature. Mechanically, these ideas can be applied to other tasks such as sequence classification. Intuitively, doing so would enable the model to attend to multiple inputs in its input context in order to better denoise the inputs. This allows the model to learn more effectively. Similar techniques can be applied to other tasks, even explored further in pretraining (Lample and Conneau, 2019).

The baseline model used here was mT5, a relatively old FLM. As a result, our baseline results are low compared to state-of-the-art NLLB results. Unfortunately, there are no better FLMs in the parameter ranges from 600M to 3B. We believe there

is still much to explore here with better FLMs, larger parameter counts, and other creative reformulations. We believe that FLMs will eventually outperform translation-only models like NLLB, due to the flexibility given by the capability to understand inputs. The input reformulations presented in this paper, which begin to bridge the performance gap between NLLB and mT5, are one example of how FLMs are more flexible in various input contexts.

## Ethics Statement

As with all work today in deep learning and large models, there are many biases introduced during large data pretraining and finetuning. We did our best to choose datasets and models which acknowledge and attempt to mitigate these biases as much as they can, and encourage the development of even better datasets and models in the future. Because the techniques introduced in this paper are input reformulations that don't introduce new data, we believe they are at least not introducing many additional risks, and are generally safe to introduce to other models and techniques. Additionally, one surprising outcome of our work is that heavy language-oriented pretraining biases were mitigated by finetuning on a language-balanced dataset. This is critical for equity with regards to multilingual applications of language models.

We believe the priority of ethics in this line of research is to ensure that the future integration of these technologies into society as safe, ethical, and trustworthy. High quality training is critical. Understanding how different inputs affect downstream performance is an important stepping stone. We encourage further research in this direction to improve model understanding and control.

Furthermore, we aim to increase accessibility of high quality, task-specific, and compute friendly large language models by improving data efficiency.

## Acknowledgements

We would like to thank Prof. Kurt Keutzer for his wisdom and hardware.

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

# A  Appendix

## A.1  Flores200 in- and out- pretrain results

Table 7: Breakdown of model and setup performance over different splits of the Flores200 dataset. "In" refers to a language that was found in the mT5 pretraining dataset and "out" refers to a language that was not. "To Eng" and "From Eng" is refererd to as xx-eng and eng-xx in some other papers, respectively. Notably, the proposed techniques improve "To Eng" performance up to 4.2 chrF++ and "From Eng" performance up to 9.4 chrF++, in the 600M case. We hypothesize this difference in improvement is due to the finetuning task including more English examples in the input, helping with downstream English translations as well as other language translations.

| Params | Setup | In-in | Out-in | In-out | Out-out | To Eng | From Eng | Avg |
|--------|-------|-------|--------|--------|---------|--------|----------|------|
| 600M | Baseline | 20.5 | 19.2 | 17.2 | 16.4 | 21.2 | 20.2 | 17.6 |
| | ParSE | 24.5 | 21.1 | 21.2 | 18.7 | 25.4 | 29.6 | 20.7 |
| | MiPS | 22.6 | 20.5 | 19.1 | 17.7 | 23.9 | 22.8 | 19.2 |
| 1B | Baseline | 28.3 | 23.6 | 17.1 | 15.2 | 33.8 | 24.6 | 20.3 |
| | ParSE | 30.9 | 25.2 | 22.7 | 19.3 | 34.6 | 32.9 | 23.8 |
| | MiPS | 27.8 | 23.6 | 19.9 | 17.7 | 31.3 | 25.8 | 21.6 |
| 3B | Baseline | 33.2 | 27.3 | 19.3 | 16.9 | 41.0 | 29.0 | 23.2 |
| | ParSE | 33.0 | 26.0 | 24.6 | 20.5 | 37.9 | 35.3 | 25.1 |
| | MiPS | 30.5 | 25.5 | 22.3 | 19.5 | 34.8 | 28.8 | 23.6 |