# OpenReview forum: "Simple and Effective Input Reformulations for Translation"
_EMNLP/2023/Conference — EMNLP 2023 Main_

### Official Review · Reviewer_Mb7f · 2023-07-26

**Soundness:** 2

**Excitement:**

2: Mediocre: This paper makes marginal contributions (vs non-contemporaneous work), so I would rather not see it in the conference.

**Paper Topic And Main Contributions:**

It is difficult to say. This paper lists many related works and reports various experiments.

**Reasons To Accept:**

This is an organized paper.

**Reasons To Reject:**

- Section 2 lists numerous related works but the idea of this work is still not clear.
- Sections 3 and 4 are "experiments on a ... task", which make the entire work an experimental report than a scientific paper.

**Reproducibility:**

3: Could reproduce the results with some difficulty. The settings of parameters are underspecified or subjectively determined; the training/evaluation data are not widely available.

**Reviewer Confidence:**

3: Pretty sure, but there's a chance I missed something. Although I have a good feel for this area in general, I did not carefully check the paper's details, e.g., the math, experimental design, or novelty.

---

> ### Author Rebuttal · Authors · 2023-08-29
>
> Thank you for reviewing our paper. We would love to respond to any specific feedback, but there was not any provided. If you could give some examples of exactly how you misunderstood the paper, we are more than happy to have a discussion about that.

---

### Official Review · Reviewer_V5ez · 2023-08-02

**Soundness:** 3

**Excitement:**

3: Ambivalent: It has merits (e.g., it reports state-of-the-art results, the idea is nice), but there are key weaknesses (e.g., it describes incremental work), and it can significantly benefit from another round of revision. However, I won't object to accepting it if my co-reviewers champion it.

**Missing References:**

-

**Paper Topic And Main Contributions:**

This paper is about the reformulation of inputs during fine-tuning for challenging translation tasks using pre-trained language models .  The authors propose three different data-efficient techniques to improve translation performance on multilingual FLMs, here is mT5, during finetuning. These techniques are the "Partial Output English Scaffold" (POSE), the "Parallel Scaffold in English" (ParSE), and the "Mixed-language Parallel Scaffold" (MiPS).
"Partial Output English Scaffold" (POSE) reformulation appends a prefix of the target English output to the input, and the "Parallel Scaffold in English" (ParSE) reformulation appends a full parallel English translation to the input. The "Mixed-language Parallel Scaffold" (MiPS) reformulation appends a different parallel translation to both the input and output for a total of 4 distinct languages per input.
The experiments are performed on the Flores200 translation benchmark and a Tibetan to English

**Questions For The Authors:**

Question A: Are these methods more suitable for low-resource languages? Does it still work for high-resource languages?
Question B: POSE works only for tibeten2eng, while ParSE works for Flores200. Which one is more generalized or in what condition to use which one?

**Reasons To Accept:**

The paper introduces three data-efficient techniques for improving translation performance during fine-tuning based on foundation language models. The paper highlights the importance of understanding how inputs affect downstream performance, which will improve model understanding and control.
The paper aims to increase accessibility to high-quality, task-specific, and compute-friendly large language models by improving data efficiency.

**Reasons To Reject:**

- The effect of the proposed methods is not very persuasive. The proposed data strategies improved the translation results by fine-tuning on foundation language models. However, there is no comparison to directly finetune the LLMs. And the best results in this paper have a big gap to the normal NMT systems.

- The paper proposed three data techniques. It is not very clear in what condition to use each one.

**Reproducibility:**

3: Could reproduce the results with some difficulty. The settings of parameters are underspecified or subjectively determined; the training/evaluation data are not widely available.

**Reviewer Confidence:**

3: Pretty sure, but there's a chance I missed something. Although I have a good feel for this area in general, I did not carefully check the paper's details, e.g., the math, experimental design, or novelty.

**Typos Grammar Style And Presentation Improvements:**

- Avoid using complex vocabulary in writing and focus on precise and accurate expression in English.

---

> ### Author Rebuttal · Authors · 2023-08-29
>
> Thank you for taking the time to thoughtfully review our paper and provide feedback. We hope our suggested edits included in the responses below demonstrate our appreciation and diligence and further demonstrate the quality of this paper.
>
> ## Response to “Paper Topic And Main Contributions” and “Reasons To Accept”
> These sections exhibit perfect understanding of the paper’s contributions. Great summary!
>
> ## Response to “Questions For The Authors”
> > “Question A: Are these methods more suitable for low-resource languages? Does it still work for high-resource languages?”
>
> From Section 4.4, we find that the balanced nature of the Flores200 dataset helps to mitigate some of the pretraining language-size bias of mT5, more equitably improving performance over low and high resource languages. Furthermore, in Section 4.4 and Figure 6, we find that the ParSE and MiPS reformulations significantly improve performance translating from higher resource in-pretrain languages across the board, up to 5.3 chrF++. These reformulations result in slight degradations translating into the same higher resource languages, up to 1.3 chrF++ on ParSE. As reported in Table 7, the proposed techniques improve high-resource English performance up to 4.2 chrF++ when translating into English and up to 9.4 chrF++ when translating from English.
>
> ***
>
> > “Question B: POSE works only for tibeten2eng, while ParSE works for Flores200. Which one is more generalized or in what condition to use which one?”
>
> Great question! As stated in the abstract and the final Introduction paragraph, we provide two separate techniques that can be applied to any translation use case. For the case of a single language pair translation task, we recommend POSE. For the case of a multi-language pair translation task, we recommend ParSE and MiPS.
>
> When this paper is accepted, the Conclusion will be improved from
> > ... We have explored how FLMs learn from their input contexts. …
>
> to
>
> > ... We have explored how FLMs learn from their input contexts. We provide two separate techniques that can be applied to any translation use case. For the case of a single language pair translation task, we recommend POSE. For the case of a multi-language pair translation task, we recommend ParSE and MiPS. …
>
> ## Response to “Reasons to Reject”
> > “However, there is no comparison to directly finetune the LLMs.”
>
> To clarify, the “baseline” condition referenced in the paper is the direct finetune condition, as described in Sections 3.2 and 4.2. Compared to this baseline direct finetune condition, our input reformulations improve performance up to 2.8 BLEU on the Classical Tibetan to English task and up to 3.5 chrF++ on the Flores200 task.
>
> ***
>
> > “And the best results in this paper have a big gap to the normal NMT systems.”
>
> This concern is addressed in the Limitations section, paragraph 2. If there were a stronger multilingual FLM to use in the parameter ranges from 600M to 3B parameters, the authors would have used them. This parameter range was selected due to hardware, compute, and accessibility constraints. Additionally, the SOTA translation model NLLB is a translation-only (non-FLM) model that our input reformulations cannot be applied to. Critically, the authors believe that the gap between translation FLMs like mT5 can reach SOTA translation-only model performance through training techniques like ParSE and MiPS. As stated in the Introduction, these techniques are specifically designed for FLMs due to their popularity and ubiquity.
>
> ***
>
> > “The paper proposed three data techniques. It is not very clear in what condition to use each one.”
>
> Please see the response to the question “Question B: POSE works only for tibeten2eng, while ParSE works for Flores200. Which one is more generalized or in what condition to use which one?”
>
> ## Response to “Typos Grammar Style And Presentation Improvements”
> > “Avoid using complex vocabulary in writing and focus on precise and accurate expression in English.”
>
> Can you please provide some concrete examples?
>
> ***
>
> Thank you again for your thoughtful comments. We hope all of your concerns were addressed here. Looking forward to hearing if there are further points we can clarify or improve.

---

### Official Review · Reviewer_wsZB · 2023-08-05

**Soundness:** 3

**Excitement:**

4: Strong: This paper deepens the understanding of some phenomenon or lowers the barriers to an existing research direction.

**Paper Topic And Main Contributions:**

This paper proposes three novel input formats for finetuning translation models. These formats adds "guidance" partial or guidance translations to the inputs, which makes it easier for the model to generate real translations. Experimental results on Flores200 and a challenging Tibenten-to-English task shows the effectiveness of the proposed approach.


**Questions For The Authors:**

1. How do you get the partial translation (in POSE) or guidance translations (in ParSE and MiPS) during inference? Do you first use the base model to generate these prefixes or guidance translations, then refine the translation by reformatting the inputs?
2. In cases of ParSE and MiPS, the choice of languages for the guidance translations can be arbitrary. How sensitive the model is to different choices of the guidance language?
3. It seems POSE can also be applied to multilingual translation tasks, and ParSE/MiPS can also be applied to the single language pair translation tasks like tib2eng. I wonder why you dedicate POSE to the single language pair case, and ParSE/MiPS to multilingual cases?
4. Are the proposed techniques more helpful in improving xx->En or En->xx translation quality?


**Reasons To Accept:**

The proposed approach is novel, simple and appears to be effective in improving MT quality.

**Reasons To Reject:**

1. Structure and writing of the paper can be improved.
2. Deeper analysis and insights is preferred.
3. Some technical details of the proposed approach can be provided.

**Reproducibility:**

4: Could mostly reproduce the results, but there may be some variation because of sample variance or minor variations in their interpretation of the protocol or method.

**Reviewer Confidence:**

4: Quite sure. I tried to check the important points carefully. It's unlikely, though conceivable, that I missed something that should affect my ratings.

**Typos Grammar Style And Presentation Improvements:**

The paper conflates methodology description and introduction in Sec. 1. It would be great to introduce the motivation and intuition in Sec. 1 first, and dedicate methodological descriptions in a separate section.

---

> ### Author Rebuttal · Authors · 2023-08-29
>
> Thank you for taking the time to thoughtfully review our paper and provide feedback. We hope our suggested edits included in the responses below demonstrate our appreciation and diligence and further demonstrate the quality of this paper.
>
>
> ## Response to “Paper Topic and Main Contributions”
> > “This paper proposes three novel input formats for finetuning translation models. These formats adds "guidance" partial or guidance translations to the inputs, which makes it easier for the model to generate real translations. Experimental results on Flores200 and a challenging Tibenten-to-English task shows the effectiveness of the proposed approach.”
>
> Great summary! To clarify, the guidance translations are applied only during finetuning and during evaluation. Additional clarification is provided in the questions responses below.
>
>
> ## Response to “Questions for the Authors”
> > “How do you get the partial translation (in POSE) or guidance translations (in ParSE and MiPS) during inference?”
>
> POSE, ParSE, and MiPS are only applied during finetuning. There is no partial or guidance translation during inference. The evaluation task consists of just the source and target translations (see Section 3.6). All setups (baseline or reformulated) use identical evaluation tasks. The main result of the paper is that the inclusion of these techniques during finetuning improves translation performance during evaluation, where there is no partial or guidance translation.
>
> During revisions, the Abstract will be improved from
> > ... These reformulations are simple data level modifications, require no additional data, and …
>
> to
>
> > ... These reformulations are simple data level modifications, require no additional training data or data at inference time, and …
>
> During revisions, the Introduction paragraph 2 will be improved from
> > ... We see qualitative improvements in the variance of the training curves, and significant increases in overall translation performance up to **10.3% / 2.8 BLEU**.
>
> to
>
> > We see qualitative improvements in the variance of the training curves. When evaluated on the same test set with no reformulations, POSE significantly increases overall translation performance compared to the direct finetuning baseline, **up to 10.3% / 2.8 BLEU**.
>
> ***
>
> > “Do you first use the base model to generate these prefixes or guidance translations, then refine the translation by reformatting the inputs?”
>
> Translation examples typically consist of a source and target translation pair. **The prefixes are simply a prefix of the target translation.** There is no prefix or guidance translation generation.
>
> During revisions, the Introduction paragraph 2 will be improved from
> > ... POSE appends a prefix of the target English output to the Classical Tibetan input. ...
>
> to
>
> > ... Tib2eng translation examples consist of a Classical Tibetan source and English target translation pair. POSE simply appends a prefix of the target English output to the Classical Tibetan input. ...
>
> During revisions, the Introduction paragraph 5 will be improved from
> > ... ParSE appends a full parallel English translation to the input. …
>
> to
>
> > ... ParSE appends the corresponding full parallel English translation (provided by Flores200) to the input. …
>
> ***
>
> > “In cases of ParSE and MiPS, the choice of languages for the guidance translations can be arbitrary. How sensitive the model is to different choices of the guidance language?”
>
> ParSE and MiPS can be seen as two extremes for choices of guidance languages. MiPS uniformly randomly samples all languages present in the Flores200 dataset, with no single dominating guidance language. MiPS improves translation performance up to 1.6 chrF++, suggesting that many choices of the guidance language would improve performance. ParSE uses English as the only guidance language. ParSE generally performed better than MiPS with improvements up to 2.8 chrF++. We conclude that using guidance languages with a larger pretraining token count correlates to better performance and the **model is somewhat sensitive to different choices of the guidance language**.
>
> ***
>
> > “It seems POSE can also be applied to multilingual translation tasks, and ParSE/MiPS can also be applied to the single language pair translation tasks like tib2eng. I wonder why you dedicate POSE to the single language pair case, and ParSE/MiPS to multilingual cases?”
>
> The motivation for POSE is discussed in Section 3.2. When finetuned on the tib2eng translation task, mT5 struggles in the initial finetuning steps. POSE is specifically designed to mitigate these struggles. In Section 4.2, we investigate POSE applied to the Flores200 task. mT5 300M achieves 16.8 chrF++ on the baseline Flores200 task, with no struggles in the initial finetuning steps, unlike the tib2eng task. Therefore, we didn't think that POSE would be successful at the Flores200 task. **Our results show that applying POSE to the Flores200 task yields 16.7 chrF++, no change to the performance, as expected.**
>
> Even though POSE specifically did not change Flores200 performance, the finetuning input reformulation idea was generalized to produce the ParSE and MiPS reformulations. These reformulations require at least 3 languages in the translation task. **As a result, the ParSE and MiPS reformulations cannot be applied to single language pair translation tasks like tib2eng.**
>
> As stated in the abstract and the final Introduction paragraph, we provide two separate techniques that can be applied to any translation use case. For the case of a single language pair translation task, we recommend POSE. For the case of a multi-language pair translation task, we recommend ParSE and MiPS.
>
> During revisions, the Conclusion will be improved from
> > ... We have explored how FLMs learn from their input contexts. …
>
> to
> > ... We have explored how FLMs learn from their input contexts. We provide two separate techniques that can be applied to any translation use case. For the case of a single language pair translation task, we recommend POSE. For the case of a multi-language pair translation task, we recommend ParSE and MiPS. …
>
> ***
>
> > “Are the proposed techniques more helpful in improving xx->En or En->xx translation quality?”
>
> As reported in Table 7 (denoted as "from Eng" and "to Eng" in the table), the proposed techniques improve xx -> En performance up to **4.2 chrF++** and En -> xx performance up to **9.4 chrF++**. The proposed techniques are more helpful in improving En -> xx translation quality.
>
> ## Response to “Typos Grammar Style And Presentation Improvements”
> > “The paper conflates methodology description and introduction in Sec. 1. It would be great to introduce the motivation and intuition in Sec. 1 first, and dedicate methodological descriptions in a separate section.”
>
> The Introduction largely consists of motivation and intuition, with select methodological descriptions of the POSE, ParSE, and MiPS reformulations to **emphasize their simplicity**. Additional methodological details are described in separate sections e.g. Sections 3.2 and 4.2. During revisions, we will consider ways to make the intuition behind the reformulations more clear in the introduction.
>
>
> ## Response to “Reasons to Reject”
> > “Deeper analysis and insights is preferred.”
>
> Section 3.7 includes analysis on finetuning loss curves and a preliminary comparison of general FLM and translation-only model performance. The POSE reformulation and the SOTA translation-only model setups can be further finetuned for increased performance.
>
> Section 4.4 includes analysis on mT5’s pretraining dataset to explain the improved performance of the ParSE and MiPS reformulations. Critically, we find that these reformulations equitably improve performance on low and high resource pretraining languages. We postulate that using a balanced training dataset like Flores200 can mitigate language frequency related biases in mT5.
>
> Section 4.4 includes analysis on encoding and decoding into different languages. We find that the ParSE and MiPS reformulations improve mT5 performance on the hardest setting of translation into languages that were not present in mT5’s pretraining language spread.
>
> ***
>
> > “Some technical details of the proposed approach can be provided.”
>
> Sections 3.1 and 4.1 include comprehensive descriptions of the finetuning and evaluation setups. Sections 3.2 and 4.2 include comprehensive descriptions of the input reformulations.
>
> ***
>
> Thank you again for your thoughtful comments! We hope all of your questions and concerns were addressed here. Looking forward to hearing if there are further points we can clarify or improve.

---

### Meta-Review · Area_Chair_Mb4C · 2023-09-17

**Recommendation:** 5

**Metareview:**

The paper introduces three data input formulation approaches aimed at enhancing translation performance during the fine-tuning process.

**Pros**:
- The proposed techniques are straightforward and consistently demonstrate effectiveness for both high-resource and low-resource languages.
- The authors conducted exhaustive experiments to understand why and when the proposed data reformulation techniques work.

**Cons**: The only drawback of this work is that when fine-tuning large foundation models, it still falls short of the performance achieved by dedicated multilingual translation models such as NLLB. Additionally, the proposed technique is not compatible for models like NLLB. Nevertheless, I believe this work retains its significance given the prevalent practice of fine-tuning language models with instructional input. The authors have adeptly addressed other queries and concerns raised by the reviewers.

---

### Decision · Program_Chairs · 2023-10-07

**Decision:**

Accept-Main

**Comment:**

The paper introduces three data input formulation approaches aimed at enhancing translation performance during the fine-tuning process.

**Pros**:
- The proposed techniques are straightforward and consistently demonstrate effectiveness for both high-resource and low-resource languages.
- The authors conducted exhaustive experiments to understand why and when the proposed data reformulation techniques work.

**Cons**: The only drawback of this work is that when fine-tuning large foundation models, it still falls short of the performance achieved by dedicated multilingual translation models such as NLLB. Additionally, the proposed technique is not compatible for models like NLLB. Nevertheless, I believe this work retains its significance given the prevalent practice of fine-tuning language models with instructional input. The authors have adeptly addressed other queries and concerns raised by the reviewers.